# Enhancing Photocatalytic Pollutant Degradation through S-Scheme Electron Transfer and Sulfur Vacancies in BiFeO₃/ZnIn₂S₄ Heterojunctions

**Ge-Ge Zheng, Xin Lin †, Zhen-Xing Wen, Yu-Hao Ding, Rui-Hui Yun, Gaurav Sharma *, Amit Kumar * and Florian J. Stadler ***

Shenzhen Key Laboratory of Polymer Science and Technology, Guangdong Research Center for Interfacial Engineering of Functional Materials, College of Materials Science and Engineering, Shenzhen University, Shenzhen 518055, China; zhenggege2020@email.szu.edu.cn (G.-G.Z.); 154679775@qq.com (X.L.); 2210342061@email.szu.edu.cn (Z.-X.W.); 2200341023@email.szu.edu.cn (Y.-H.D.); yuanruihui2022@email.szu.edu.cn (R.-H.Y.)

* Correspondence: gaurav8777@gmail.com (G.S.); mittuchem83@gmail.com (A.K.); fjstadler@szu.edu.cn (F.J.S.)

† Current address: Foxconn Engineering Services Division, iLab Precision Testing and Failure Analysis Department, Shenzhen 518110, China.

**Abstract:** Photocatalytic degradation plays a crucial role in wastewater treatment, and the key to achieving high efficiency is to develop photocatalytic systems that possess excellent light absorption, carrier separation efficiency, and surface-active sites. Among various photocatalytic systems, S-type heterojunctions have shown remarkable potential for efficient degradation. This work delves into the construction of S-type heterojunctions of ternary indium metal sulfide and bismuth ferrite nanofibers with the introduction of sulfur vacancy defects and morphology modifications to enhance the photocatalytic degradation performance. Through the impregnation method, BiFeO₃/ZnIn₂S₄ heterojunction materials were synthesized and optimized. The 30% BiFeO₃/ZnIn₂S₄ heterojunction exhibited superior photocatalytic performance with higher sulfur vacancy concentration than ZnIn₂S₄. The in-situ XPS results demonstrate that the electrons between ZnIn₂S₄ and BFO are transferred via the S-Scheme, and after modification, ZnIn₂S₄ has a more favorable surface morphology for electron transport, and its flower-like structure interacts with the nanofibers of BFO, which has a further enhancement of the reaction efficiency for degrading pollutants. This exceptional material demonstrated a remarkable 99% degradation of Evans blue within 45 min and a significant 68% degradation of ciprofloxacin within 90 min. This work provides a feasible idea for developing photocatalysts to deal with the problem of polluted water resources under practical conditions.

**Keywords:** indium zinc sulfide; bismuth ferrate nanofibers; S-type heterojunctions; sulfur vacancies; photogenerated carrier efficiency; photocatalytic degradation

## 1. Introduction

With the rapid advancement of industry and technology, the ecological environment of our planet is under enormous strain due to the extensive consumption of non-renewable energy sources, such as coal, oil, and natural gas. In addition, the discharge of organic pollutants, such as organic dyes, nitroaromatic compounds, and fertilizer waste, into industrial wastewater, has caused significant damage to our water bodies and poses a serious threat to human health [1,2]. Finding sustainable solutions to environmental pollution is a paramount challenge and a significant area of scientific research.

Advanced oxidation processes (AOPs), which involve the generation of reactive oxygen species (ROS) to degrade organic compounds, represent an effective technique widely studied for environmental remediation [3]. Semiconductor photocatalysis, as an AOP, has emerged as one of the most promising strategies for addressing contemporary environmental problems, owing to its mild reaction conditions, high efficiency, environmental

friendliness, and low cost [4–6]. It is capable of directly converting solar energy into the photocatalytic synthesis of organic compounds, photocatalytic decomposition of water, photocatalytic reduction of carbon dioxide to produce hydrocarbon fuels, photocatalytic degradation of pollutants, and photocatalytic sterilization [7].

However, current narrow-bandgap photocatalysts, such as titanium dioxide ($TiO_2$), have limitations, as they require only UV light excitation, which accounts for only 2.4% of sunlight [4,8]. Consequently, researchers have been exploring narrow-band gap photocatalysts, such as $MoS_2$ [9] and CdS [10], while also focusing on modifying single photocatalysts to improve their catalytic activity. Composites and heterojunction structures have been widely studied as effective ways to address these limitations [11].

This study focuses on the photocatalytic degradation of biological dyes and antibiotic drugs using a ternary metal sulfide composite bismuth ferrite photocatalyst.

Ternary sulfur compounds, a novel class of semiconductor photocatalysts, have recently garnered significant attention in the field of photocatalysis due to their narrow bandgap, high stability, and strong visible light absorption properties [12,13]. In comparison to metal oxides, these compounds possess a wider forbidden band and exhibit a broader range of light response, resulting in superior carrier transport behavior [14]. Notwithstanding, conventional ternary metal sulfides have been plagued by poor stability. Thus, researchers have begun to shift their focus towards ternary metal sulfides, such as $ZnIn_2S_4$ [15], $CdIn_2S_4$ [16], $CuInS_2$ [17], and $AgIn_5S_8$ [18], which have been shown to offer richer active sites and improved stability as compared to monometallic sulfides [19,20]. Although the reduction efficiency of these compounds may be limited by the slow electron transfer behavior and high carrier recombination efficiency, strategies such as morphology control, defect engineering, heterostructure construction, and co-catalyst loading have been explored to enhance their interface structure and behavior, ultimately leading to improved conversion efficiency and selectivity [21,22].

Zinc indium sulfide ($ZnIn_2S_4$), a widely studied ternary metal sulfide, exhibits two different polycrystalline forms, hexagonal and cubic, depending on the synthesis method. In the hexagonal $ZnIn_2S_4$ crystal structure, atoms are stacked along the c-axis in a repeating sequence of S-Zn-S-In-S-In-S, with Zn and half of the In atoms tetrahedrally coordinated by S atoms, and the remaining In atoms octahedrally coordinated [15,23]. In cubic $ZnIn_2S_4$, Zn atoms are tetrahedrally coordinated by S atoms, while In atoms are octahedrally coordinated. The photocatalytic performance of $ZnIn_2S_4$ is highly dependent on its structure, morphology, and optical properties [24]. Hence, morphology modification has been extensively applied, where the introduction of anionic vacancies in the semiconductor can not only enhance its light absorption ability, but also introduce mid-gap states in the bandgap that serve as effective electron "traps", promoting the separation and transfer efficiency of photogenerated carriers [25–27]. Moreover, the composition of a heterojunction by combining two or more photocatalysts can regulate the flow direction of photogenerated carriers and improve their utilization efficiency, ultimately leading to enhanced photocatalytic degradation performance of the semiconductor photocatalyst. In recent years, the construction of S-type heterojunctions has emerged as a promising goal for the development of new photocatalytic heterojunction systems [28,29].

Bismuth ferrite, or $BiFeO_3$, is a perovskite-type photocatalytic material that has garnered attention in the field due to its narrow band gap of 2.1 eV, good chemical stability, and remarkable multiferroic and piezoelectric properties. However, the photocatalytic performance of $BiFeO_3$ is hampered by its high recombination rate, short lifetime, and poor light absorption of photo-generated carriers [30–32]. Interestingly, given its band structure compatibility with $ZnIn_2S_4$, the current study has employed a microwave hydrothermal method to synthesize **s**ulfur **v**acancy-modified (VS) $ZnIn_2S_4$ and an electrospinning technique to prepare $BiFeO_3$ nanofibers, followed by the composite preparation of different ratios of $BiFeO_3/ZnIn_2S_4$ binary heterojunctions through an impregnation method. Various characterizations have been conducted to establish the carrier transfer path of $BiFeO_3/ZnIn_2S_4$ binary heterojunctions under light illumination, as well as to elucidate the photocatalytic

performance enhancement arising from its S-type band structure. The photocatalytic degradation of Evans Blue dye and enrofloxacin antibiotic served as benchmarks to measure the efficiency of $BiFeO_3/ZnIn_2S_4$ binary heterojunctions.

## 2. Materials and Methods

### 2.1. Materials

Zn(CH₃COO)₂ (AR), thioacetamide (TAA, 99%), N,N-dimethylformamide (DMF, AR), and CH₃COOH (AR, 99.5%) were obtained from Macklin (Shanghai, China). Bi(NO₃)₃·5H₂O (≥98%), Fe(NO₃)₃·9H₂O (AR) was obtained from Aladdin, and InCl₃ (>99%) was obtained from TCI Shanghai. All the materials were used without further purification. Deionized water was used throughout the experiments.

#### 2.1.1. Synthesis of Pure BiFeO₃

A solution was prepared by dissolving 1.94 g of Bi(NO₃)₃·5H₂O and 1.616 g of Fe(NO₃)₃·9H₂O in a mixture of 8 mL of DMF and 2 mL of acetic acid. The solution was stirred continuously until it became transparent. Then, 1.4 g of PVP (with a molecular weight $M_w$ of 1,300,000 g/mol) was slowly added to the solution, which was stirred overnight to create a precursor solution (Figure 1).

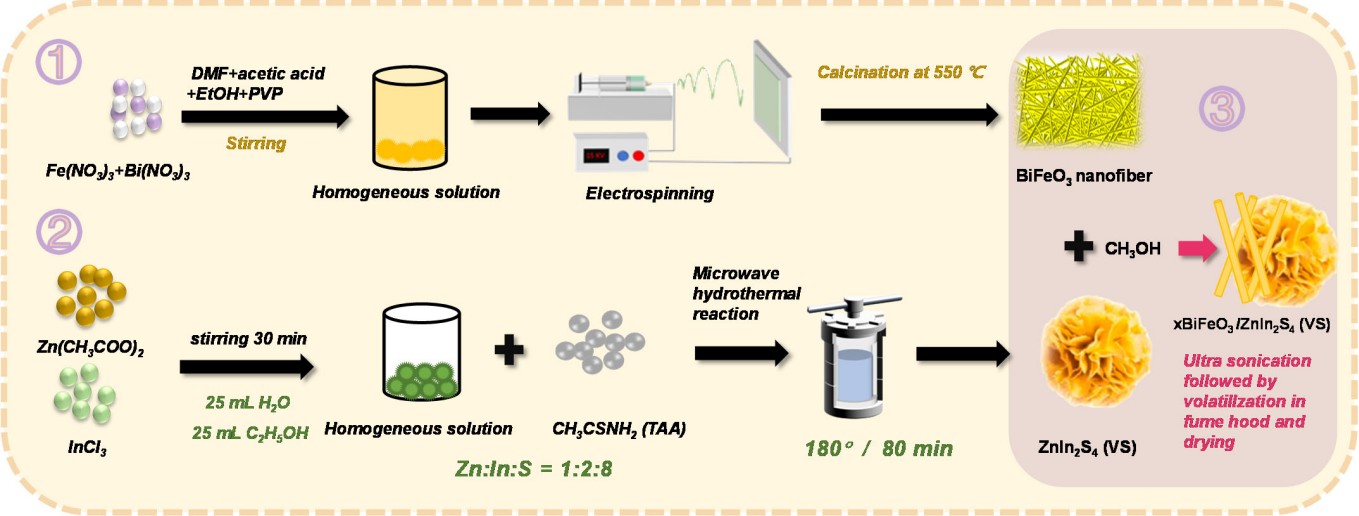

**Figure 1.** Synthesis scheme for $BiFeO_3/ZnIn_2S_4$ (BFO/ZISS).

The precursor solution was loaded into a 10 mL syringe equipped with a stainless-steel needle with an inner diameter of 0.41 mm. The voltages applied to the tip and the collection roller were 15 kV and −1 kV, respectively, while the distance between the tip and the collector was maintained at 15 cm. The feed rate was set at 0.3 milliliters per hour, and the humidity level was controlled to be around 50% RH.

The precursor nanofibers collected needed to be quickly transferred to an oven set at 60 °C, where they were dried overnight to volatilize the solvent. Subsequently, the precursor nanofibers were placed in a muffle furnace and calcined at 550 °C for two hours at a heating rate of 5°/min. This process yielded the final BiFeO₃ nanofibers.

#### 2.1.2. Synthesis of Pure ZnIn₂S₄

To initiate the synthesis of the desired compound, a 1:1 aqueous ethanol solution was employed to dissolve 0.4 mmol of Zn(CH₃COO)₂ and 0.8 mmol of InCl₃. Following 30 min of stirring, 3.2 mmol of TAA was added to the solution after ensuring that it had achieved homogeneity, followed by continued stirring until complete dissolution was observed. Next, the solution was transferred onto a Teflon reaction substrate, heated at 180 °C for 80 min and allowed to cool to room temperature. The resulting product was washed

multiple times with deionized water and ethanol and was then dried using a vacuum oven at a temperature of 50 °C for a duration of 12 h (Figure 1).

### 2.1.3. Synthesis of $BiFeO_3/ZnIn_2S_4$ Heterojunction

The present study employed the wet-impregnation method to prepare $BiFeO_3/ZnIn_2S_4$ heterojunctions with varying ratios. Specifically, $BiFeO_3$ and $ZnIn_2S_4$ were separately sonicated and added to 30 mL of methanol. The resulting $BiFeO_3$ suspension and $ZnIn_2S_4$ suspension were mixed and stirred until the methanol solution had completely evaporated. The resulting composite powder samples were then thoroughly washed with water and ethanol, and they were finally dried in an oven at 60 °C for 12 h. Composite samples in the proportions of 20%, 30%, and 40% were prepared, while pure $BiFeO_3$ and $ZnIn_2S_4$ (VS) were respectively labeled as BFO and ZISS. The resulting composite samples were labeled as 20BFO/ZISS, 30BFO/ZISS, and 40BFO/ZISS (Figure 1).

### 2.1.4. Simulation of the Degradation of Contaminated Water

Evans Blue is a disazo dye. It has strong water solubility and a high affinity for serum albumin, making it suitable for protein tracing analysis, histology, and fluorescence microscopy. Its azo structure and strong water solubility make it difficult to remove from polluted water. In this experiment, a 20 mg/L concentration of Evans Blue solution was prepared to simulate the dye concentration in polluted water. In each photocatalytic degradation experiment, 20 mg of the photocatalyst was used to degrade 100 mL of Evans blue dye.

Ciprofloxacin (1-cyclopropyl-6-fluoro-7-(1-piperazinyl)-1,4-dihydro-4-oxo-quinoline-3-carboxylic acid) belongs to the third generation of fluoroquinolone antibiotics and is currently one of the most widely used fluoroquinolone antibiotics in the world. Its strong benzene ring structure makes it difficult to degrade. In this experiment, a 20 mg/L concentration of ciprofloxacin suspension was prepared as the target pollutant to analyze the degradation performance of the photocatalyst on antibiotics. In each photocatalytic degradation experiment, 20 mg of the photocatalyst was used to degrade 100 mL of ciprofloxacin suspension.

### 2.2. Characterization

X-ray diffraction (XRD, SmartLab, Rigaku, Tokyo, Japan) and X-ray photoelectron spectroscopy (XPS, K-Alpha+, ThermoFisher Scientific, Waltham, MA, USA), were used to determine the composition and structure of the material, while scanning electron microscopy (SEM, SU-70, HITACHI, Tokyo, Japan) was used to observe its surface morphology. XRD involved scanning the sample at room temperature using Cu K$\alpha$-radiation, while XPS involved measuring the number of photoelectrons of different energies to characterize the change in electron density and the direction of charge flow after the photocatalytic reaction. SEM detected secondary and backscattered electrons to convert physical signals into image information, with a gold coating treatment used for poorly conductive or non-conductive samples. The instrument was operated at 5 kV voltage, at a distance of 10 mm, and different magnifications were used to observe the microscopic morphology of the material. An energy dispersive spectroscopy (EDX, X Max, Oxford Instruments, London, UK) was also used to analyze the distribution of elements in the sample.

Transmission Electron Microscope (TEM) was used to determine the microstructure and internal crystal structure of samples using an F200 (JEOL, Akishima, Tokyo, Japan) at 80 kV. HRTEM (High Resolution TEM) images provided lattice patterns, and interplanar spacing was characterized using Image J software. Samples were prepared by dispersing them in anhydrous ethanol and dropping them onto a copper grid.

UV-vis spectroscopy was used to test the absorption spectrum of Evans blue and enrofloxacin in a quartz cuvette. The UV-vis Spectrophotometer (UV-2450, SHIMADZU, Kyoto, Japan) was used to obtain the concentration of target degradation products by

comparing the absorbance spectrum, which was used to calculate the photocatalytic degradation rate.

DRS tested the absorbance intensity of solid samples using barium sulfate as a reference material at 200–800 nm. The semiconductor material's $E_g$ bandgap width was obtained using the formula [33]:

$$(\alpha h v)^{1/n} = A(hv - E_g), \tag{1}$$

where; $\alpha$ is the absorption coefficient, $A$ is the transition constant, and $hv$ is the energy of incident photons. The $n$ value is $1/2$ for direct bandgap semiconductors and 2 for indirect bandgap semiconductors. The bandgap width was obtained by extrapolating the straight line part of the graph to the abscissa axis.

The technique of steady-state photoluminescence spectroscopy (PL) is employed to ascertain the recombination efficiency of electron-hole pairs in a semiconductor material. Sample preparation and the use of (FS5, Edinburgh, UK) are key aspects of this approach. Another method used is electron paramagnetic resonance (EPR/ESR), which uses an ESR-spectrometer to detect free radicals generated in a system. The addition of 5,5-diethyl-1-pyrroline-N-oxide (DMPO) as a spin-trapping agent aids in determining the main active groups of the photocatalyst during the reaction. Electrochemical impedance spectroscopy (EIS), Mott-Schottky (M-S) measurements, and transient photocurrent (i-t) measurements were conducted using a CHI760 (Shanghai Chenhua Technology, Shanghai, China) to evaluate the transport efficiency and resistance of charge transport, to determine the flat band potential of a semiconductor material, and to study the separation and transfer efficiency of photogenerated charge carriers under light/dark conditions. Sample preparation is also important in these methods. The total organic carbon content was measured using a Total Organic Carbon Analyzer (Multi N/C 2100, Analytik Jena AG, Jena, Germany) to estimate the removal of the organic matter from the samples to check for the efficiency of the breakdown of the pollutants beyond the first step, which is important to ensure that potentially toxic intermediate degradation products are eliminated, as well as the initial pollutant.

The experimental setup in this study was a photocatalytic reactor utilizing a 300 W Xenon lamp and reflux water to regulate the temperature during the photocatalytic degradation of the target pollutants. The reactor can accommodate up to 8 simultaneous photocatalytic reactions. The synthesized photocatalyst was tested for its photocatalytic performance by degrading the colored dye Evans Blue (EB) and the colorless antibiotic drug Ciprofloxacin (CIP) under simulated visible light. To carry out the experiment, 20 mg photocatalyst in 100 mL of target degradation solution were placed in the reactor, and after a 30-min dark reaction, the light source was turned on for 90 min. Samples were collected every 15 min, and pollutant concentrations were determined using UV-Vis spectroscopy.

The absorbance of the target degradate solution was obtained by UV-Vis spectrophotometer test for different light exposure times, and according to the Lambert-Beer law equation, for the same intensity of incident light, the light absorbance (i.e., absorbance) of the colored solution is proportional to the product of solution concentration and liquid layer thickness:

$$A = \lg\frac{1}{T} = Kbc \tag{2}$$

The aforementioned formula illustrates that the absorbance of a solution (A) is directly proportional to the concentration (c) and path length (b) of the absorbance layer, with a constant (K) and the transmittance (T). By tracking the change in absorbance of the target degradation solution at various illumination times, the change in concentration of the time-dependent pollutant concentration can be computed using the following formula:

$$\% \text{ degradation} = (A_0 - A)/A_0 \times 100\% = (C_0 - C)/C_0 \times 100\% \tag{3}$$

Here, $A_0$ and $C_0$ represent the initial absorbance and concentration of the target pollutant, while A and C denote the absorbance and concentration of the pollutant at different reaction times.

## 3. Results

The XRD patterns of ZISS, BFO, 20BFO/ZISS, 30BFO/ZISS, and 40BFO/ZISS are displayed in Figure 2. For pure ZISS, the data corresponds to the characteristic diffraction peaks at $2\theta = 21.59°$, $27.69°$, $30.45°$, $39.78°$, $47.18°$, $52.44°$, and $55.58°$, indexed to the crystal planes (006), (102), (104), (108), (110), (116), and (022) of the orthorhombic square crystal system $ZnIn_2S_4$ (JCPDS No. 65-2023) [34,35], respectively. For pure BFO, the characteristic diffraction peaks at $2\theta = 22.39°$, $31.73°$, $32.01°$, $39.41°$, $45.69°$, $50.28°$, and $56.87°$ correspond to the (012), (104), (110), (202), (024), (211), and (214) planes of hexagonal $BiFeO_3$ (JCPDS No. 71-2494) [36]. The characteristic diffraction peaks for different ratios of BFO/ZISS heterojunction show the characteristic peaks of each of the two materials without the presence of any impurity peaks, indicating that the BFO/ZISS heterojunctions were successfully synthesized. In addition, the intensity ratio of the characteristic peaks of BFO gradually increases with the increase in the BFO content, indicating the successful synthesis of different ratios of BFO/ZISS heterojunctions.

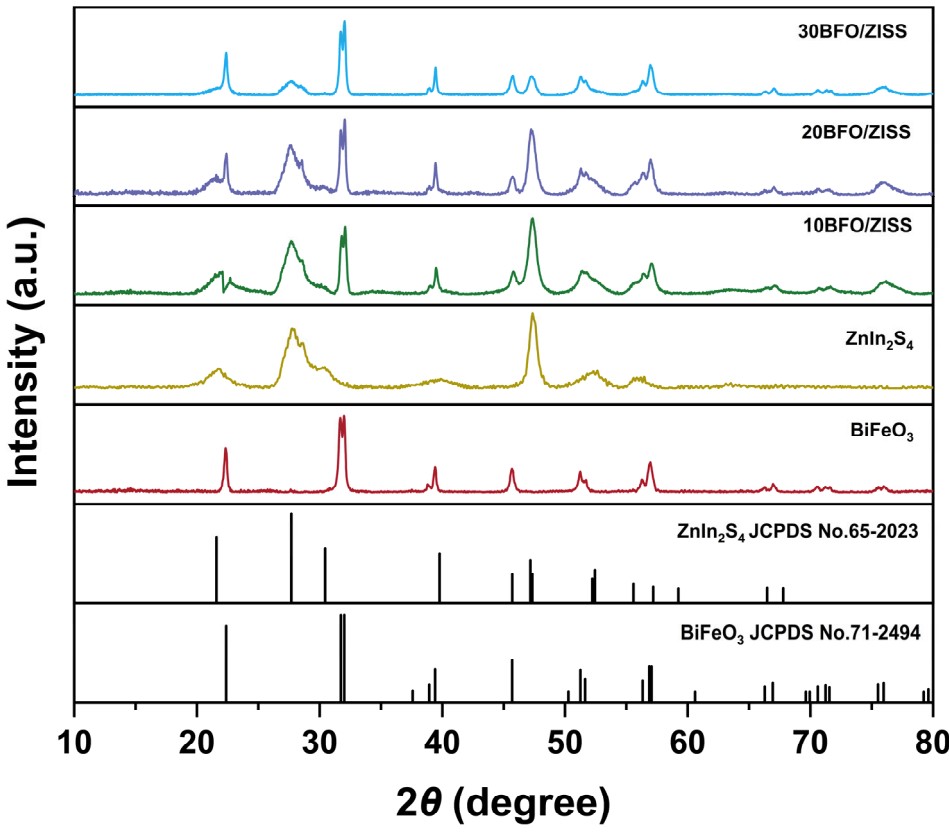

**Figure 2.** XRD-data of ZIS, BFO, 20BFO/ZISS, 30BFO/ZISS, and 40BFO/ZISS along with the JCPDS-reference data.

The surface chemical composition of the BFO/ZISS heterojunction has been subject to thorough XPS analysis, and the results are presented herein. The survey spectrum of the 30BFO/ZISS sample exhibits the presence of all elements (Bi, S, In, O, Fe, Zn) in the sample, as displayed in Figure 3a. This observation confirms the presence of the element In in the form of $In^{2+}$ [37]. The binding energy spectrum of Zn 2p for pure ZISS, displayed in Figure 3b, is deconvoluted into two peaks at 1019.9 eV and 1045.0 eV, corresponding to Zn $2p_{3/2}$ and Zn $2p_{1/2}$, respectively, thereby indicating the presence of $Zn^{2+}$ [37] in the crystal structure. In Figure 3c, the high-resolution XPS spectra of In 3d are shown,

with two main peaks at 444.9 eV and 452.5 eV, corresponding to In $3d_{5/2}$ and In $3d_{3/2}$, respectively. Upon formation of the BFO/ZISS heterojunction, a slight positive chemical shift in B.E. is observed, suggesting that electrons shift from ZISS to BFO. Moreover, after light irradiation, the binding energy of Zn 2p shifts to the negative direction, indicating that the electrons transfer from BFO to ZISS. The shifts can also be seen in the Bi 4f-S 2p spectrogram (Figure 3e), where two peaks of S 2p are observed at 161.7 eV and 162.9 eV due to the spin–orbit separation of the S element, indicating that the S element is present as $S^{2-}$ [38]. For pure BFO, the peaks observed at 157.6 eV and 158.8 eV belong to Bi $4f_{7/2}$ and at 162.9 eV and 164.1 eV belong to Bi $4f_{5/2}$, indicating the presence of $Bi^{3+}$ [39] in the BFO nanofibers. In the BFO/ZISS heterojunction, the negative binding energy of Bi $4f_{7/2}$ is observed, demonstrating that BFO receives electrons from ZISS when the two semiconductors are in contact. On light exposure, the binding energy of Bi $4f_{7/2}$ shifts positively, indicating that the transfer of electrons flows from BFO to ZISS during the photocatalytic reaction. As shown in Figure 3d, the three peaks of O 1s with binding energies of 529.5 eV, 531.3 eV, and 533.3 eV correspond to lattice oxygen, oxygen adsorbed on the surface, and oxygen in water [40], respectively. The Fe $2p_{3/2}$ peak positions of 711.4 eV and 709.8 eV for $Fe^{3+}$ and $Fe^{2+}$, respectively, and the Fe $2p_{1/2}$ peak positions of 726.1 eV and 723.5 eV for $Fe^{3+}$ and $Fe^{2+}$ in the pure BFO, respectively, accompanied by the presence of two satellite peaks at 718.3 eV and 731.6 eV, demonstrate the $Fe^{3+}$ oxidation state (Figure 3f). After the formation of the BFO/ZISS heterojunction, the $2p_{3/2}$ and Fe $2p_{1/2}$ of Fe 2p could not continue to deconvolute the peak, indicating that $Fe^{2+}$ was oxidized to $Fe^{3+}$ by the electrons transferred from ZISS [41,42], which is consistent with the chemical shift phenomenon occurring in the Zn and Bi elements.

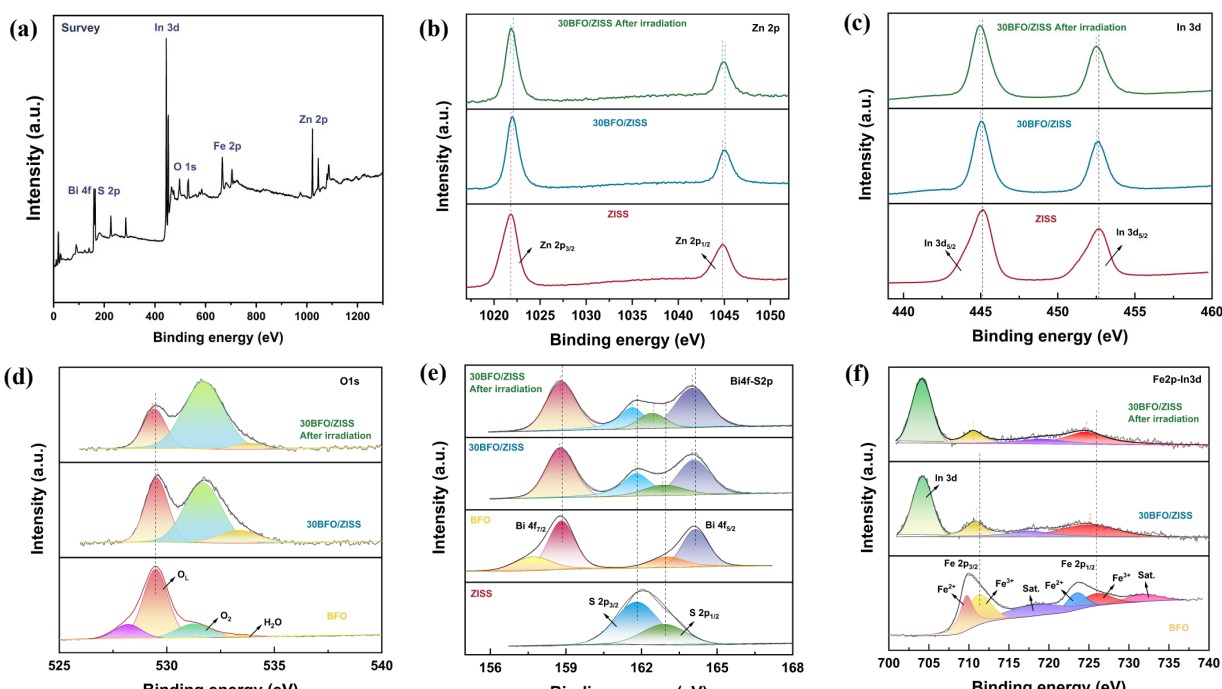

**Figure 3.** XPS-data, (**a**) survey spectrum, (**b**) In 3d, (**c**) Zn 2p, (**d**) O1s, (**e**) Bi4f-S2p, (**f**) Fe2p-In3d.

The observation of the shifts in binding energy (B.E) following the contact of BFO and ZISS unequivocally establishes the transfer of electrons from ZISS to BFO. Such a phenomenon is attributed to the differences in Fermi levels, as ZISS (with a value of 4.59 eV) [43], exhibits a lower work function than BFO (with a value of 4.7 eV) [44]. This process persists until the Fermi levels have reached equilibrium, giving rise to an internal electric field that results in the bending of bands. Following light exposure, the photogenerated electrons migrate from the conduction band (CB) of BFO to the valence

band (VB) of ZISS and combine via a S-scheme transfer. This conclusion is corroborated by the shifts observed in the binding energy (B.E) in the XPS spectra.

The existence of vacancy defects in the samples was additionally corroborated through EPR tests. As depicted in Figure 4, robust EPR signals were observed at g = 2.01 for both ZISS and 30BFO/ZISS, attesting to the presence of S vacancies in both ZISS and 30BFO/ZISS. In comparison to ZISS, the composite heterojunction 30BFO/ZISS exhibited a more intense S vacancy signal, suggesting that 30BFO/ZISS has a higher concentration of S vacancies [45]. The higher concentration of S vacancies in 30BFO/ZISS, which serves as an active site for trapping photoinduced electrons, is favorable for suppressing the recombination of photogenerated carriers and extending the lifetime of photogenerated carriers, thereby enhancing photocatalytic performance [46,47].

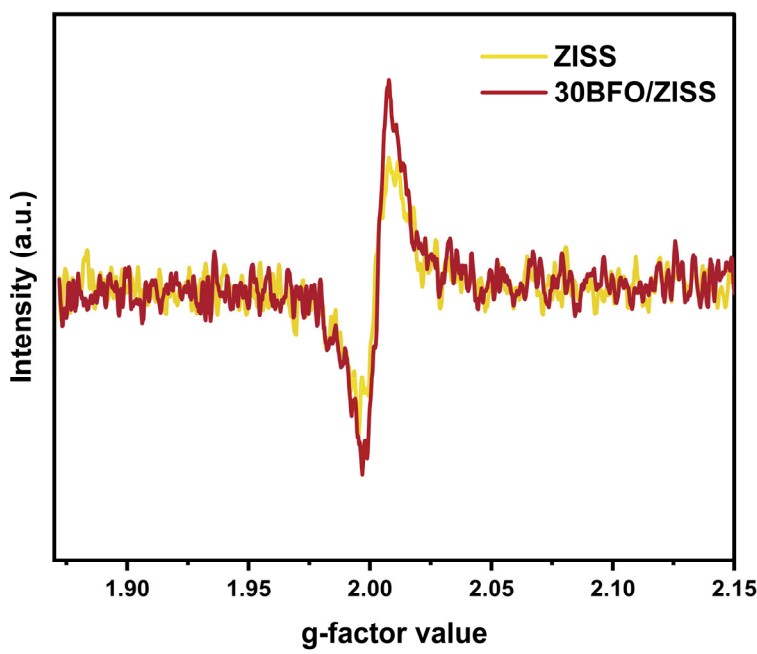

**Figure 4.** EPR spectra of ZISS and BFO/ZISS composite heterojunctions.

In order to analyze the morphology and microstructure of the prepared photocatalysts, SEM, TEM, and EDS analysis were performed. As shown in Figure 5a,b, the ZISS nanoflower structure is composed of multilayer sheets. As shown in Figure 5c,d, the synthesized BFO nanofibers have a uniform fiber distribution and no fracture, and the fibers can be seen to have a diameter of about 50–200 nm and a high aspect ratio, providing more reactive sites for photocatalytic reaction. Figure 5e shows that, in the BFO/ZISS heterojunction, the photocatalyst fibrous BFO and the lamellar composition of the nanoflower ZISS closely adhere to each other, and both BFO and ZISS maintain their original morphology after heterojunction formation, and the closely adhered interface is beneficial to the formation of heterojunction and charge transfer. As shown in Figures 5f and S1, the constituent elements Zn, In, S, Fe, Bi, and O are uniformly distributed in the composite sample, indicating that the BFO/ZISS heterojunction was successfully synthesized.

The characterization of the BFO/ZISS heterojunction was carried out using TEM and HRTEM techniques to analyze the microscopic morphology and phase composition. In the low-resolution TEM images (Figure 6a,b), clear observations of BFO nanofibers and ZISS nanoflowers composed of flakes were made. Additionally, high-resolution TEM (Figure 6c) revealed lattice spacings of 0.27 nm and 0.32 nm, corresponding to the (110) crystal plane of BFO (JCPDS No.71-2494, darker subcrystal) and the (102) crystal plane of ZISS (JCPDS No. 65-2023, lighter subcrystal), respectively. HRTEM provided evidence of the tight contact interface between BFO and ZISS, characterizing the formation of heterojunctions.

The lattice stripes shown in HRTEM corresponded to XRD standard cards, confirming the successful synthesis of the BFO/ZISS composite heterojunctions.

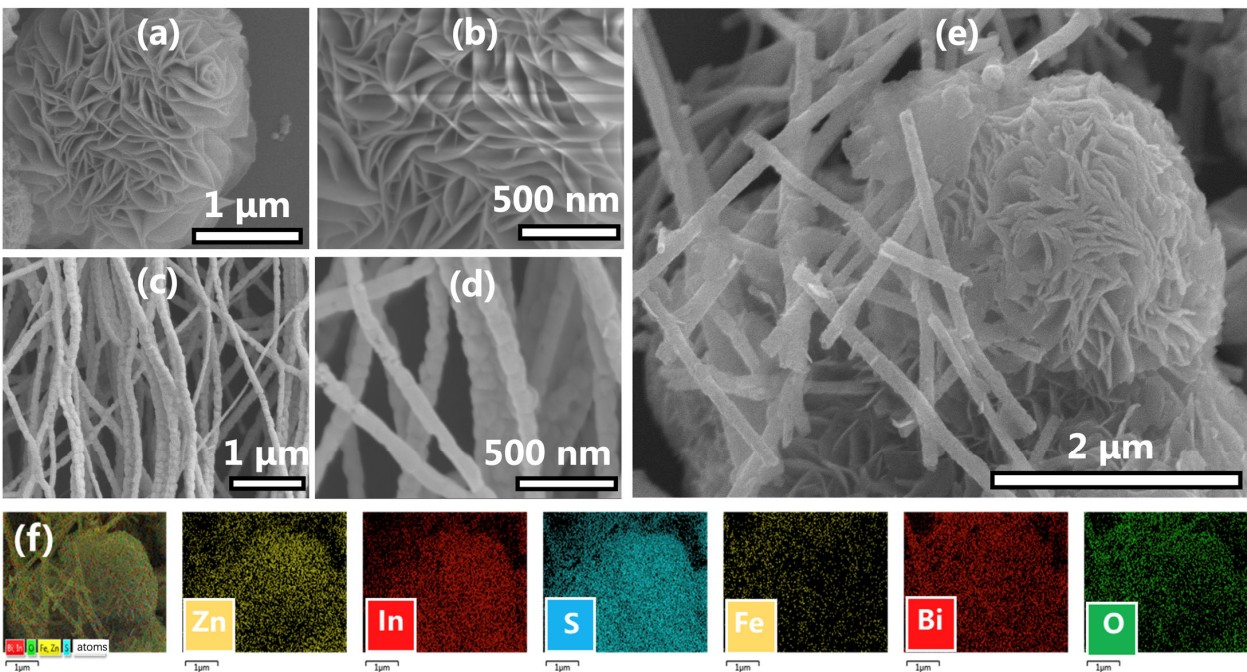

**Figure 5.** (**a**,**b**) SEM images of ZISS; (**c**,**d**) SEM images of BFO; (**e**) SEM images of BFO/ZISS; (**f**) SEM images of BFO/ZISS EDS diagram.

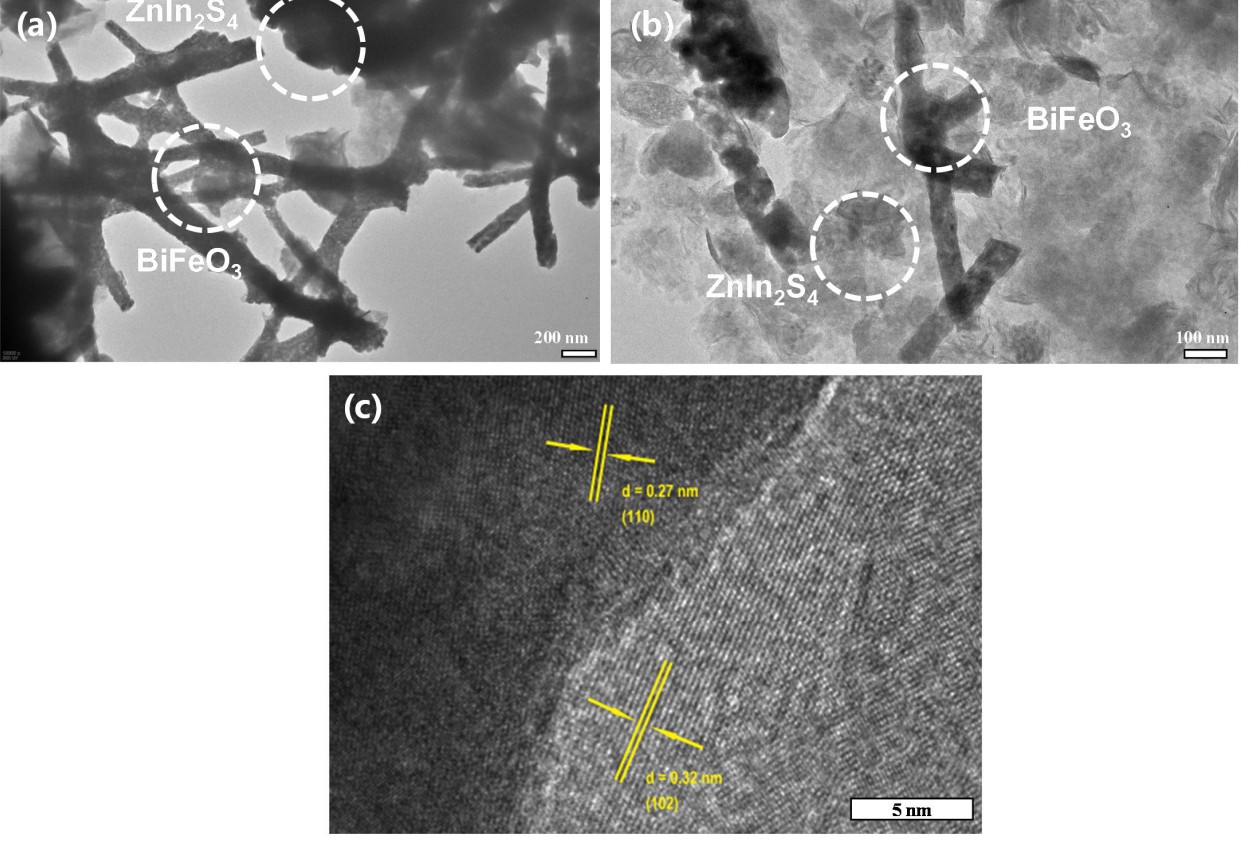

**Figure 6.** (**a**,**b**) TEM images; (**c**) HRTEM image of BFO/ZISS.

UV-DRS spectra were used to study the light absorption properties of photocatalysts. Figure 7a shows DRS plots for different ratios of BFO/ZISS, as well as pure BFO and pure ZISS. All materials have a high light absorption capacity in the UV range (200~400 nm) due to the strong absorption of UV light by pure ZISS. As compared to ZIS, BFO shows higher visible absorption owing to its lower band gap, and the BFO/ZISS composite heterojunction has a higher absorption of visible light than the pure components. This is due to incorporation of BFO in the junction and, as well as, due to sulfur vacancies. The optimized 30BFO/ZISS sample showed the highest intensity of visible light absorption.

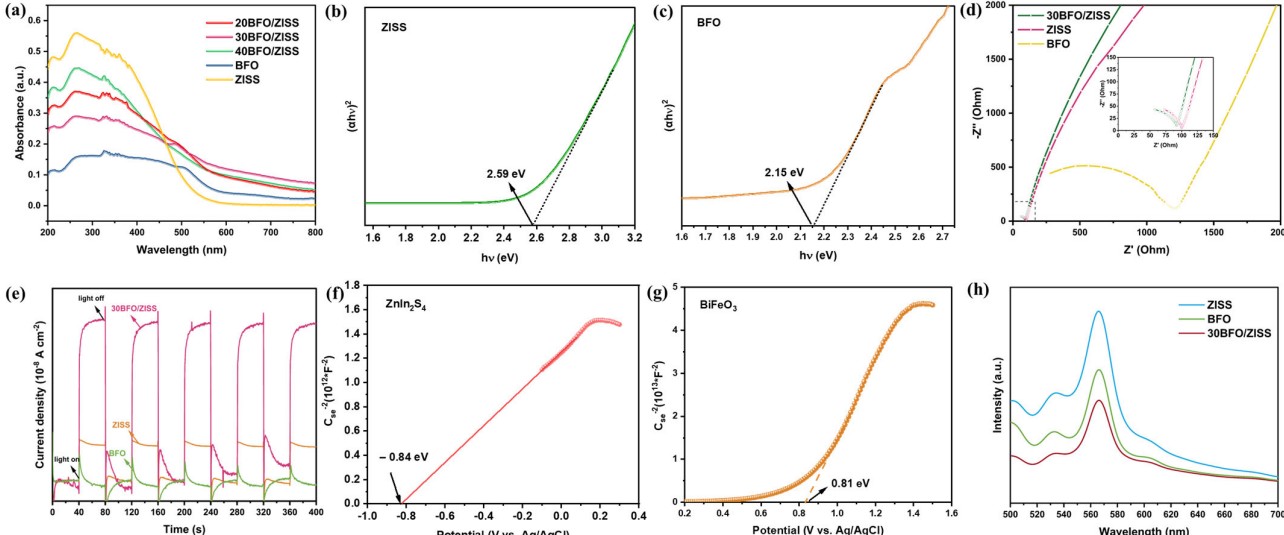

**Figure 7.** (**a**) UV-DRS-data, Mott-Schottky plots of (**b**) ZISS and (**c**) BFO, (**d**) EIS-data, (**e**) photocurrent response, (**f**,**g**) M-S plots of ZISS and BFO (**h**) photoluminescence spectra of BFO, ZISS, and 30BFO/ZISS.

To calculate the band gap energy, the Kubelka-Munk formula was used, which showed the band gap energy of ZISS and BFO to be 2.59 eV and 2.15 eV, respectively (Figure 7b,c). As shown in the Figure 7f,g, the flat-band potential $E_{FB}$ was calculated using Mott-Schottky plots, which showed that the $E_{FB}$ of ZISS was −0.84 V (vs. Ag/AgCl) or −0.64 V (vs. NHE), while the $E_{FB}$ of BFO was 0.81 V (vs. Ag/AgCl) or 1.01 V (vs. NHE).

The M-S curve for both ZISS and BFO had a positive slope, indicating that they are both n-type semiconductors (Figure 7b,c). The conduction band potential of a n-type semiconductor is typically lower than the flat band potential by 0.1–0.2 V (vs. NHE). By combining the results from the Kubelka-Munk formula and the Mott-Schottky plots, the $E_{VB}$ of ZISS was calculated to be 1.75 V vs. NHE, while the EVB of BFO was 2.96 V vs. NHE.

To further investigate the charge transfer behavior of the photocatalysts, electrochemical impedance tests (EIS) were performed on the ZISS, BFO, and 30BFO/ZISS heterojunction, as shown in Figure 7d. From the EIS plots, the Nyquist radius of the samples was observed, and it is clear that 30BFO/ZISS has the smallest arc radius, which means that it has better charge transfer capacity by the formation of heterojunction. This is due to effective S-scheme charge transfer facilitated by an internal electric field. The photocurrent densities of ZISS, BFO, and 30BFO/ZISS are shown in Figure 7e, and, according to the EIS results, the heterojunctions have the highest photocurrent response, which can be attributed to the higher visible light absorption of 30BFO/ZISS, as well as the high charge transfer rate and the modulation of the electron-hole pair separation by the heterojunction, which allows the photogenerated carriers to have a longer lifetime. In addition, the 30BFO/ZISS maintained stable photocurrent intensity after five switching cycles, indicating that the heterojunction photocatalyst has good photocurrent stability.

Figure 7h shows the photoluminescence spectra of BFO, ZISS, and 30BFO/ZISS photocatalysts. The stronger the PL intensity, the faster the recombination of light-generated

electron–hole pairs. The poor PL intensity for 30BFO/ZISS is due to the formation of a heterojunction with S-type energy band structure, which effectively separates the electron–hole pairs and reduces the recombination of electron–hole pairs. In addition, the presence of sulfur vacancies acts as surface-active points to prolong the photogenerated carrier lifetime. The formation of S-type heterojunctions and the presence of S-hole synergistically improve the efficiency of photogenerated carriers.

The degradation efficacy of a BFO/ZISS heterojunction photocatalyst was assessed via the degradation of two target pollutants, namely, Evans blue (EB) dye and ciprofloxacin (CIP) antibiotic. Figure 8 displays the photocatalytic performance of ZISS, BFO, and three distinct ratios of BFO/ZISS photocatalysts in the degradation of EB dye at pH = 1, 2, 5, 7, and 10. As depicted in Figure 8a, the first thirty minutes are the dark reaction without illumination, during which EB molecules undergo adsorption–desorption reactions with the photocatalysts. It is discernible that 40BFO/ZISS exhibits the most favorable adsorption performance. As the illumination progresses, the photocatalytic degradation of EB by 30BFO/ZISS is accomplished within 45 min, achieving a degradation rate of 99%, whereas the degradation rate of 40BFO/ZISS amounts to 93%. It can be perceived that, as the BFO proportion increases from 20% to 40%, the degradation rate of the heterojunction photocatalyst initially escalates and then descends. The sample 30BFO/ZISS evinces the highest degradation rate, concurrent with the characterization results of EIS, PL, and photocurrent response, thereby indicating that 30BFO/ZISS embodies the best composite ratio. Figure 8b illustrates the corresponding pseudo-first-order kinetic constants, and it can be observed that the reaction rate of 30BFO/ZISS ($0.13508$ min$^{-1}$) is the highest, exceeding that of ZISS by 1.7 times and BFO by 26.9 times. We compare the other studies with this work on the degradation of Evans Blue dye on Table S1 and found a good performance of our photocatalytic system in comparison to the literature results [48–54].

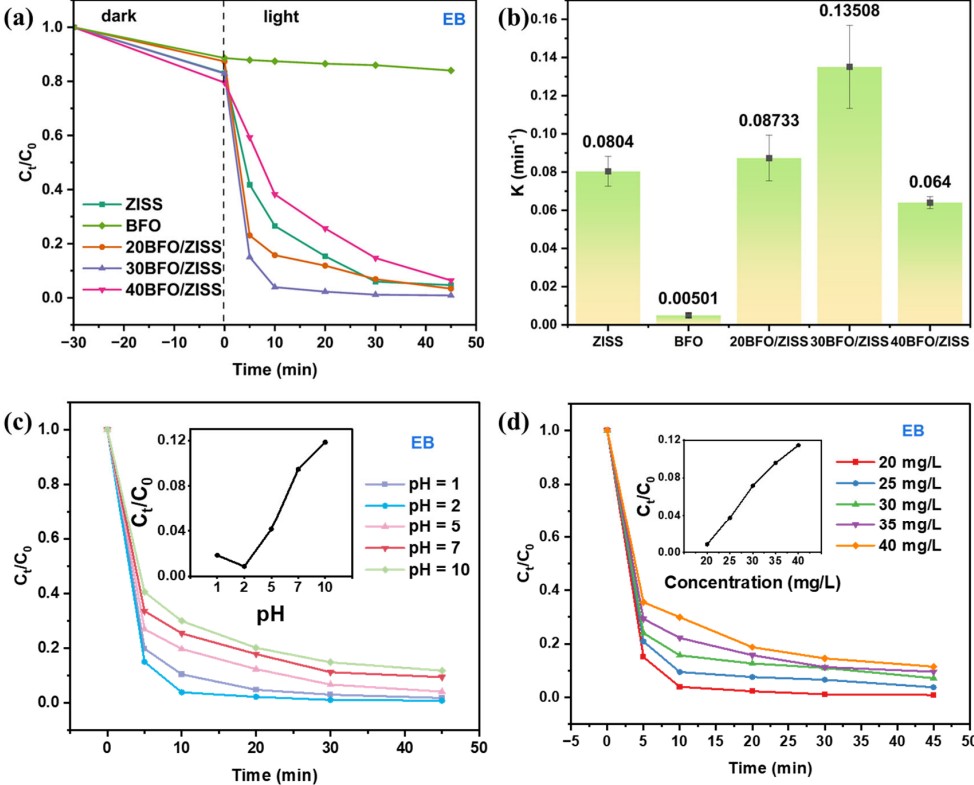

**Figure 8.** Photocatalytic performance against Evans Blue (EB). (**a**) concentration of EB vs. time for different photocatalysts, (**b**) resulting kinetic constants, (**c**) pH-dependence, and (**d**) EB–concentration dependence.

Furthermore, the effect of various pH environments on the photocatalytic degradation efficacy of 30BFO/ZISS in the degradation of EB was scrutinized. As illustrated in Figure 8c,

as the pH decreases, the degradation rates of 30BFO/ZISS for EB within 45 min are 89%, 91%, 96%, 99%, and 98%, respectively, implying that 30BFO/ZISS performs well in both acidic and alkaline environments. Notably, 30BFO/ZISS displays the best photocatalytic degradation performance at pH = 2.

Figure 8d depicts the photocatalytic degradation performance of 30BFO/ZISS for different concentrations of EB dye. It is evident that, as the pollutant concentration escalates from 20 mg/L to 40 mg/L, 30BFO/ZISS exhibits a high degradation rate. Even at a high concentration of 40 mg/L, the degradation rate reaches 89% within 45 min, indicating that 30BFO/ZISS still demonstrates good photocatalytic performance even in a high concentration of dye environment. Owing to its higher sulfur vacancy concentration, 30BFO/ZISS can adsorb a significant number of dye molecules as a surface-active site, thus exhibiting better catalytic degradation performance than ZISS for high-concentration pollutants.

Figure 9 illustrates the photocatalytic performance of various composite photocatalysts, including BFO, ZISS, and three different BFO/ZISS composites, for the degradation of CIP antibiotics. The experimental results indicate that 30BFO/ZISS exhibits superior photocatalytic degradation performance compared to BFO and ZISS at pH = 2, 7, 9, 10, and 12 (Figure 9a). Specifically, the removal of CIP by 30BFO/ZISS after 30 min of reaction in the dark is 16%, which is higher than that of BFO and ZISS. In the presence of light, 30BFO/ZISS demonstrates the best degradation performance, achieving a degradation of 68% within 90 min, which is significantly higher than that of ZISS (43%) and BFO (9%). The degradation performance of 30BFO/ZISS on CIP was further improved compared to other work, Table S2, and it found a good performance of our photocatalytic system in comparison to the literature results [55–59].

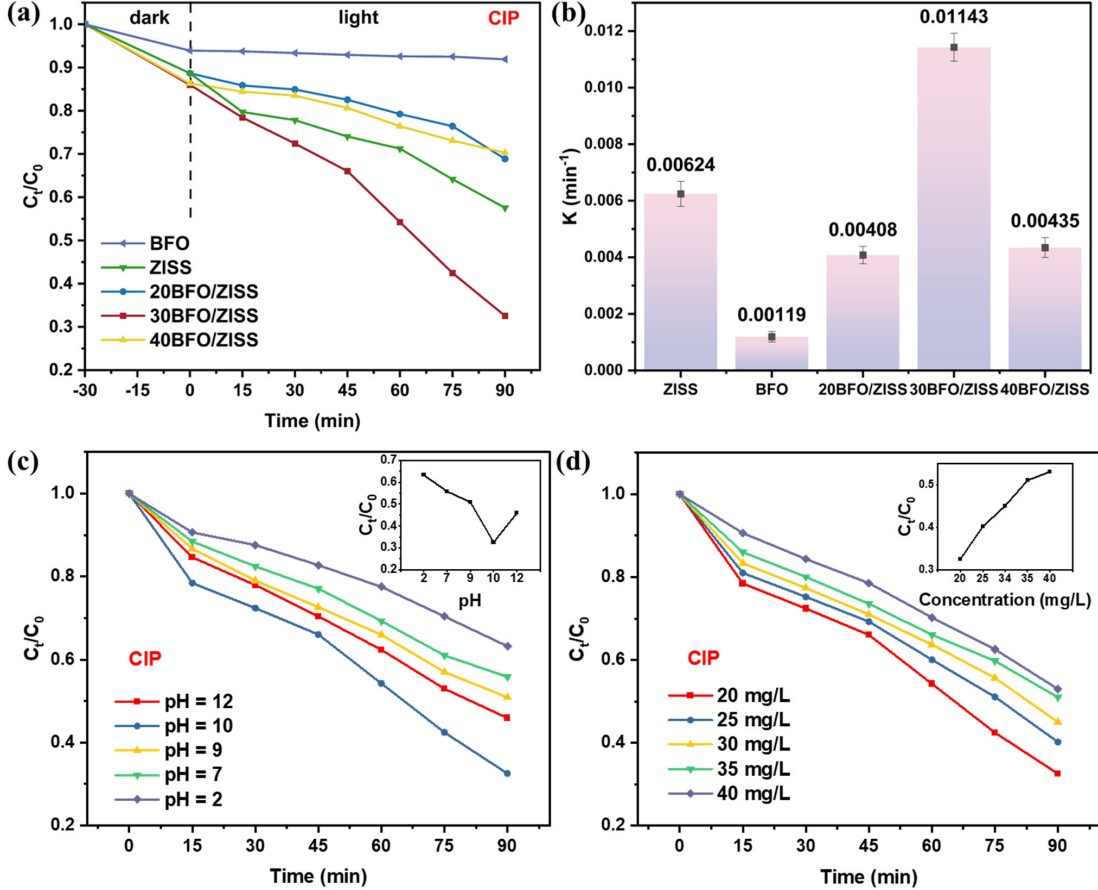

**Figure 9.** Photocatalytic performance against ciprofloxacin (CIP). (**a**) concentration of CIP vs. time for different photocatalysts, (**b**) resulting kinetic constants, (**c**) pH dependence, and (**d**) CIP–concentration dependence.

The enhancement in photocatalytic degradation performance can be attributed to the synergistic effect of BFO and ZISS, which improves the adsorption performance of the photocatalyst. Moreover, the concentration of surface defects in 30BFO/ZISS increases, indicating that sulfur vacancies act as surface active sites, capture more electrons, and adsorb more pollutants, achieving higher degradation performance. The kinetic constants (Figure 9b) also demonstrate that 30BFO/ZISS has the highest degradation rate ($0.01143$ min$^{-1}$).

The effect of different pH values of CIP solutions on the photocatalytic degradation performance of 30BFO/ZISS is also investigated (Figure 9c). It is observed that, as the pH decreases from 12 to 2, the degradation rate of CIP by 30BFO/ZISS initially increases and then decreases, reaching a maximum at pH = 10, which is 68%. Under strongly acidic conditions, the degradation rate decreases to 37%, indicating that acidic conditions are not conducive to the degradation of CIP by 30BFO/ZISS photocatalyst. This could be due to the limiting effect of hydrogen ions on the adsorption of antibiotic molecules by the photocatalyst, thereby restricting the photocatalytic reaction.

Furthermore, the effect of different concentrations of CIP on the photocatalytic activity of 30BFO/ZISS is investigated (Figure 9d). The results show that the degradation rate of CIP decreases with increasing concentrations of CIP, and, at a concentration of 40 mg/L, the degradation rate decreases to 48%. However, even at a high concentration of 40 mg/L, the degradation rate of 30BFO/ZISS is still higher than that of ZISS at a concentration of 20 mg/L, indicating the superior performance of the 30BFO/ZISS composite photocatalyst. The reason for the decrease in degradation rate may be due to the accumulation of high concentrations of CIP molecules on the surface of the photocatalyst, which leads to low light utilization efficiency and, thus, reduces the photocatalytic performance.

In addition, we also tested the photocatalytic degradation performance of 30BFO/ZISS for malachite green, which is an azo dye with Evans Blue, and norfloxacin, which is a quinolone antibiotic with ciprofloxacin. Figure S2a,b illustrates the degradation performance of Evans Blue compared to Malachite Green (pH = 2, c(photocatalyst) = 20 mg/100 mL, c(pollutant) = 20 mg/L, T = 25 °C) and the antibiotics giprofloxacin and norfloxacin, respectively, under identical conditions (pH = 10, c(photocatalyst) = 20 mg/100 mL, c(pollutant) = 20 mg/L, T = 25 °C). The better degradation performance against the same type of dyes and antibiotics indicates that 30BFO/ZISS can effectively degrade dyes and antibiotic contaminants in the real environment to some extent. Figure S3 shows the total organic carbon removal of the two pollutants after 60 min of photocatalytic degradation for Evans Blue and ciprofloxacin, respectively. It is obvious that the catalysts remove the organic carbon intermediates of both pollutants, which suggests a good photodegradation efficiency for not only the base molecule, but also for the organic molecules, which are created from these pollutants.

ESR spectroscopy was employed to identify the primary reactive species involved in the photocatalytic degradation of the dye EB on the 30BFO/ZISS composite material. Upon visible light irradiation, a high-intensity signal of $\cdot O_2^-$ was detected with a peak intensity ratio of 1:1:1:1, while no signal was detected in the dark, indicating that $\cdot O_2^-$ is the main active species responsible for the degradation process (Figure 10a). The ESR spectrum of $\cdot OH$ displayed a lower peak intensity ratio of 1:2:2:1, indicating a minor role in the degradation process (Figure 10b). The generation of these species is elucidated by the energy band structure of the heterojunction. The 30BFO/ZISS composite possesses sufficient potential to generate $\cdot O_2^-$, owing to the more negative conduction band potential of ZISS and the higher valence band potential of BFO. In comparison to ZISS, the composite structure provides adequate valence band potential to stimulate the generation of $\cdot OH$, thereby augmenting the photocatalytic activity of the material.

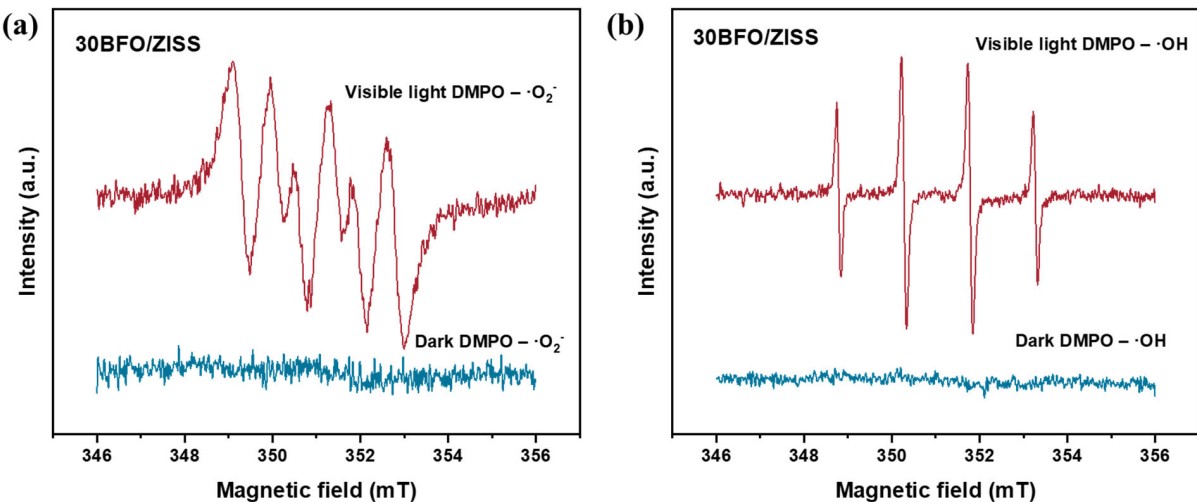

**Figure 10.** ESR-spectroscopy. (**a**) $\cdot O_2{}^-$, (**b**) $\cdot OH$.

The stability of a photocatalyst has a profound impact on its practical deployment. As illustrated in Figure 11, the SEM and XRD images of 30BFO/ZISS after five cycles of use provide evidence of the sample's microstructure, comprising ZISS nanoflowers and fibrous BFO, which appears to be sufficiently stable and not significantly altered. Moreover, the XRD outcomes reveal that the peak intensity and position of 30BFO/ZISS's characteristic peaks remained unaltered following repeated utilization, which confirms the crystal structure's sustained stability in the composite photocatalyst. These findings thus suggest that 30BFO/ZISS possesses favorable durability and reusability attributes, which bodes well for its pragmatic utility.

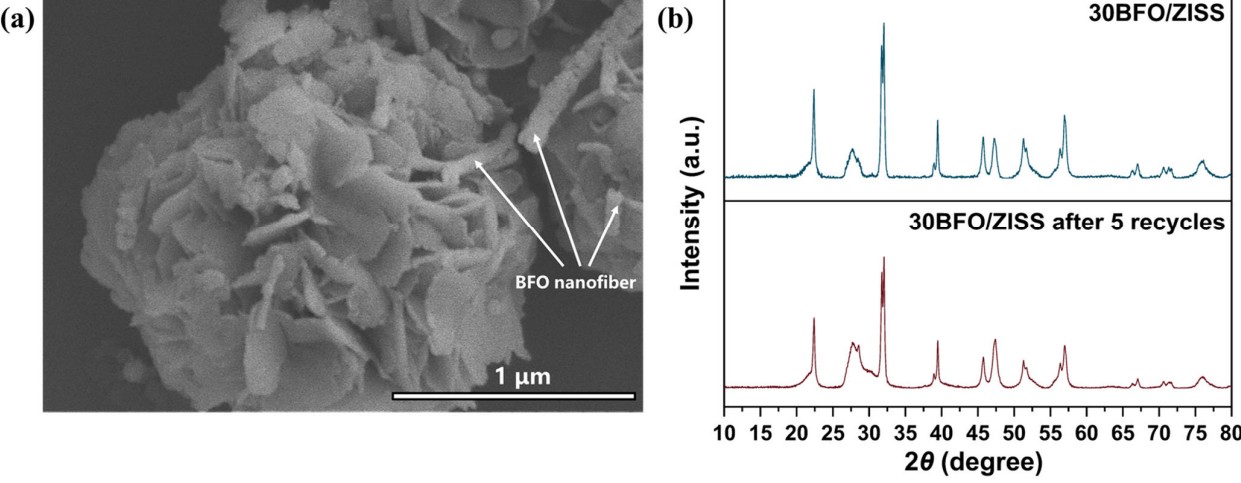

**Figure 11.** (**a**) SEM of 30BFO/ZISS after five cycles of use and (**b**) XRD images before and after five cycles of use.

The comparison of the XPS spectra of 30BFO/ZISS heterojunction photocatalyst before and after photocatalytic reaction is shown in Figure S4. It can be inferred that no significant intensity changes and large peak position shifts are observed. In addition, no substantial change in oxidation states of Fe and In are observed. Minor changes in binding energies are obviously due to light exposure, which causes a shift in electrons among the semiconductors. This is also observed in in situ XPS results discussed earlier. Thus, it is deduced that the surface composition of the photocatalyst was stable.

The energy band structures of BFO and ZISS were derived via UV-DRS and M-S testing, as depicted in Figure 12. BFO is an exemplary oxidative photocatalyst (OPs) with

a higher work function and lower Fermi level [47]. In contrast, ZISS resembles typical reductive photocatalysts(RPs) with a lower work function and higher Fermi level [60]. As a result of ZISS's smaller work function, electrons have the propensity to migrate from ZISS to BFO when in contact, until the Fermi levels of both semiconductors reach equilibrium. The contact interface of BFO becomes negatively charged due to electron accumulation, causing the energy band to bend downwards. Simultaneously, ZISS becomes positively charged due to electron loss, causing the energy band to bend upwards. An internal built-in electric field, directed towards BFO, is established at the heterojunction. Additionally, ZISS produces defect energy levels (VS) due to the presence of **s**ulfur **v**acancies [61].

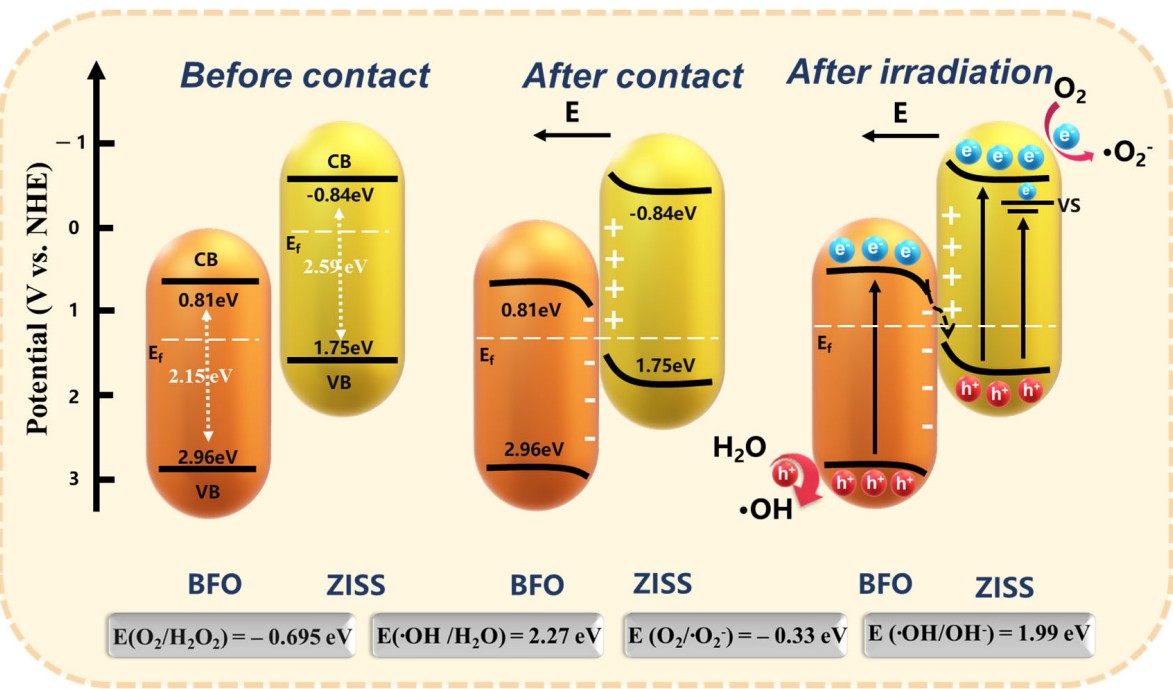

**Figure 12.** Schematic of the obtained S-type heterojunction and the resulting energy levels.

When subjected to light, the BFO/ZISS semiconductor heterojunction generates photo-generated electron–hole pairs. Owing to the bending of the energy band, the formation of an internal built-in electric field, and the Coulomb force, electrons on BFO's conduction band (CB) and holes on ZISS's valence band (VB) recombine, whereas electrons on ZISS's CB and holes on BFO's VB are retained by the bent energy band. The interface charge transfer behavior of the BFO/ZISS heterojunction is congruent with the XPS test findings, signifying the formation of a S-type heterojunction. Moreover, owing to the ample oxidation–reduction potential of ZISS's CB and BFO's VB, $\cdot O_2{}^-$, $\cdot OH$, free radicals, and photo-generated electron–hole pairs participate in the reaction, resulting in the BFO/ZISS heterojunction's superior photocatalytic degradation performance. The S-scheme transfer has already been validated by the in-situ XPS measurements, as discussed in previous sections. The sulfur vacancies are usually located under the bottom of CB, and they provide active sites and favorable metallic conductivity. The S vacancies are not only responsible for improved visible light absorption, but they also behave as active trapping sites for electrons and, thereby, further improve the charge carrier's separation. Thus, both the S-scheme charge transfer and abundant S-vacancies synergistically improve the charge transfer, light absorption capacity, charge separation, and strong redox capability.

## 4. Conclusions

In summary, we successfully synthesized $BiFeO_3/ZnIn_2S_4$ composite S-type heterojunctions of binary nanoflowers/nanofibers by the wet impregnation method and investigated their photocatalytic degradation performance against Evans Blue dye and

ciprofloxacin antibiotics. The composite ratio of $BiFeO_3/ZnIn_2S_4$ was optimized, and the composite with 30% $BiFeO_3/ZnIn_2S_4$ ratio was obtained to have the best degradation performance. 30BFO/ZISS showed excellent photocatalytic activity for both Evans Blue and ciprofloxacin by simulating visible light for photocatalytic degradation experiments, and the degradation of Evans Blue within 45 min is 99% and 68% degradation of ciprofloxacin in 90 min. The efficient photocatalytic performance of 30BFO/ZISS was attributed to the high visible light absorption enhanced by defect and morphology modification, better charge transfer capability, longer photogenerated carrier lifetime, effective S-scheme electron transfer mechanism, and excellent photo-redox ability. From the EIS, PL, photocurrent response, and energy band structure analysis, it was inferred that the best performing photocatalyst 30BFO/ZISS had excellent redox ability, high charge separation, electron transfer ability, and excellent photocurrent response. In addition, XPS and band structure analysis confirmed that the charge transfer in 30BFO/ZISS heterojunctions is in accordance with the S-transfer mechanism, and EPR results showed that the composite 30BFO/ZISS heterojunctions have a stronger S vacancy signal, which plays an important role in enhancing charge utilization as a surface-active site. Evans dye represents a challenging-to-degrade disazo dye, while the ciprofloxacin antibiotic represents a widely used fluoroquinolone antibiotic with a representative structure and degradation difficulty. Therefore, the excellent degradation performance and structural stability in cyclic tests of 30BFO/ZISS for the degradation of Evans Blue dye and ciprofloxacin antibiotics makes it a promising candidate for practical applications.

**Supplementary Materials:** The following supporting information can be downloaded at: https://www.mdpi.com/article/10.3390/jcs7070280/s1, Table S1: different photocatalyst and their performances to degrade Evans blue, Table S2: $ZnIn_2S_4$ based photocatalyst and their performances to degrade Ciprofloxacin, Figure S1: The EDX spectrum elemental composition, Figure S2: 30BFO/ZISS photocatalytic performance, (a) Comparison of azo dyes; (b) Comparison of quinolone antibiotics, Figure S3: TOC removal for Evans Blue and Ciprofloxacin degradation by 30BFO/ZISS, Figure S4: XPS spectra after photocatalytic degradation reaction, (a) survey spectrum, (b) Zn 2p, (c) In 3d, (d) O 1s, (e) Bi 4f-S 2p, (f) Fe 2p-In 3d.

**Author Contributions:** Conceptualization, F.J.S.; Photocatalysis experiments, G.-G.Z. and X.L.; Sample preparation, G.-G.Z. and X.L.; Morphological characterization experiments, G.-G.Z., F.J.S. and R.-H.Y.; data processing analysis, G.-G.Z., A.K., G.S. and Z.-X.W.; writing—original draft preparation, G.-G.Z., A.K., G.S. and F.J.S.; writing—review and editing, F.J.S., G.-G.Z., A.K., G.S. and Y.-H.D.; supervision, Funding acquisition: F.J.S.; project administration, F.J.S. All authors have read and agreed to the published version of the manuscript.

**Funding:** The authors wish to acknowledge funding from the National Science Foundation of China (5221101262) and from Shenzhen City (JCYJ20210324093205013).

**Data Availability Statement:** Data are available upon reasonable request from the authors.

**Acknowledgments:** The authors thank the School of Materials Science and Engineering and the central laboratory, Shenzhen University for the SEM, TEM, XPS, XRD, EDS, and FTIR-characterization. This work was created based on the first author's MSc-thesis using ChatGPT (versions from April/May 2023) to translate the text from Chinese and polish the language.

**Conflicts of Interest:** The authors declare no conflict of interest.

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
