# Peer review of "Enhancing Photocatalytic Pollutant Degradation through S-Scheme Electron Transfer and Sulfur Vacancies in BiFeO3/ZnIn2S4 Heterojunctions"

_jcs, doi:10.3390/jcs7070280_

Round 1
Reviewer 1 Report
The text presents a well-structured and coherent description of the research conducted on the synthesis and optimization of BiFeO3/ZnIn2S4 composite S-type heterojunctions for efficient photocatalytic degradation of pollutants.
The text could also provide more information on the specific pollutants that were tested in the photocatalytic degradation experiments, as well as the concentration and volume of the pollutants used (in Materials and Methods).
Finally, the authors could discuss the potential practical applications of their findings, such as the potential for scaling up the synthesis of the composite photocatalysts for industrial wastewater treatment.
It would be beneficial to include references or comparisons with other similar studies.
Line 171, reference this equation.
Line 201: Authors write: "...and the transmittance (T)." but in the formula in line 181 the transmittance does not appear, please clarify.
Line 240: Here appears the abbreviation ZIS but also ZISS, what is the difference between them. In Figure 2 I don't find ZIS.
Line 311: nanoflower structure composed = nanoflower structure is composed
Line: 335, name the abbreviation "M-S plots", please added.
The text presents a well-structured and coherent description of the research conducted on the synthesis and optimization of BiFeO3/ZnIn2S4 composite S-type heterojunctions for efficient photocatalytic degradation of pollutants.
The text could also provide more information on the specific pollutants that were tested in the photocatalytic degradation experiments, as well as the concentration and volume of the pollutants used (in Materials and Methods).
Finally, the authors could discuss the potential practical applications of their findings, such as the potential for scaling up the synthesis of the composite photocatalysts for industrial wastewater treatment.
It would be beneficial to include references or comparisons with other similar studies.
Line 171, reference this equation.
Line 201: Authors write: "...and the transmittance (T)." but in the formula in line 181 the transmittance does not appear, please clarify.
Line 240: Here appears the abbreviation ZIS but also ZISS, what is the difference between them. In Figure 2 I don't find ZIS.
Line 311: nanoflower structure composed = nanoflower structure is composed
Line: 335, name the abbreviation "M-S plots", please added.
Reviewer 2 Report
The current manuscript discussed "Enhancing Photocatalytic Pollutant Degradation through S-2 Scheme Electron Transfer and Sulfur Vacancies in 3 BiFeO3/ZnIn2S4 Heterojunctions formation by chemical and hydrothermal methods. Authors explained the fabrication of BiFeO3/ZnIn2S4 heterojunctions and characterization via various techniques. The photocatalytic properties were studied for selected photodegradation materials like Evans blue and ciprofloxacin. The article is well written and properly structured however, there are some concerns which needs to be addressed before considering the manuscript for publication.
There are some comments:
1. The authors should add a table and compare different ferrite materials reported in literature used for similar purpose and explain how their heterojunction is novel and better.
2. The selection of Evans blue and ciprofloxacin was made on what bases, why not similar class of materials were selected for better comparison.? It is recommended to use at least one pollutant of similar class either from Evans blue or from ciprofloxacin and present a better comparison.
3. The composition of BiFeO3/ZnIn2S4 should be clearly mentioned. It is not clear which component clearly enhances the photocatalytic activity of the material, Authors should describe, why a particular composition was successful for photodegradation.
4. The authors should also provide XRD and XPS of post photodegradation samples to compare the differences in composition and oxidation states before and after the photo degradation experiments.
5. The EDX elemental composition should be mentioned in table form.
6. It will be of great interest to provide the TOC analysis of the photocatalysis process, as mentioned that BiFeO3/ZnIn2S4 heterojunction is a potential material for studying degradation of organic dye and pharmaceutical pollutants.
Minor editing, rephrasing is required.
Round 2
Reviewer 1 Report
The authors have considered my comments. This paper could be accepted for publication.
Author Response
thank you, no changes done based on your comments.
Reviewer 2 Report
The authors have addressed all the concerns properly .
Author Response

(The authors gave the same response as above.)
